# Weighted Geometric Mean (WGM) method: A new chromatic adaptation model

**Che Shen**[ID][☾]*, **Mark D. Fairchild**[☾]

Munsell Color Science Laboratory, Rochester Institute of Technology, Rochester, New York, United States of America

☾ These authors contributed equally to this work.
* cs7607@g.rit.edu

**Data Availability Statement:** All relevant data are within the manuscript and its Supporting Information files. Derby/Leeds data:https://doi.org/10.1002/(SICI)1520-6378(199908)24:4%3C295::AID-COL10%3E3.0.CO;2-K Cai data:http://dx.doi.org/10.1002/col.22228. Shen data:http://dx.doi.org/10.2352/issn.2169-2629.2021.29.374 Fairchild

## Abstract

The human visual system has undergone evolutionary changes to develop sophisticated mechanisms that enable stable color perception under varying illumination. These mechanisms are known as chromatic adaptation, a fundamental aspect of color vision. Chromatic adaptation can be divided into two categories: sensory adaptation, which involves automatic adjustments in the visual system, such as retinal gain control, in response to changes in the stimulus, and cognitive adaptation, which depends on the observer's knowledge of the scene and context. The geometric mean has been suggested to be the fundamental mathematical relationship that governs peripheral sensory adaptation. This paper proposes the WGM model, an advanced chromatic adaptation model based on a weighted geometric mean approach that can anticipate incomplete adaptation as it moves along the Planckian or Daylight locus. Compared with two other chromatic adaptation models (CAT16 and vK20), the WGM model is tested with different corresponding color data sets and found to be a significantly improvement while also predicting degree of adaptation (sensory and cognitive adaptation) in a physiologically plausible manner.

## Introduction

Chromatic adaptation refers the ability of the human visual system to adapt (completely or incompletely) to the prevailing illumination in order to approximately maintain the perceived object colors across a range of ambient light conditions. This is the most important aspect of the human visual system to understand and model color appearance. Luo mentioned that various chromatic adaptation transforms (CATs) have been derived from fitting a corresponding colors data sets, and the majority of the CATs include three steps of calculation as illustrated in Fig 1 and detailed below [1].

Step 1: Any physiologically plausible CAT model must act on signals approximating the cone responses (LMS) that can be accurately converted from CIE tristimulus values (XYZ) via a linear 3x3 matrix, such as $M_{16}$ [2].

Reversibility data:https://doi.org/10.2352/CIC.2022.
30.1.28.

**Funding:** The author(s) received no specific
funding for this work.

**Competing interests:** The authors have declared
that no competing interests exist.

Step 2: This step converts the cone responses (LMS), under the test illuminant ($L_nM_nS_n$), into
the adapted cone responses ($L_cM_cS_c$) by using the CAT's defined mathematical transforma-
tion, often based on von Kries scaling of the cone responses [3].

Step 3: Finally, the adapted cone responses ($L_cM_cS_c$) are transformed back to CIE tristimulus
values using the inverse of the 3x3 matrix used in step 1. This enables practical colorimetric
applications that are often standardized around CIE XYZ Tristimulus values.

For more than a century, Step 2, or the CAT itself, has been investigated by many color
scientists. In 1902, von Kries published his hypothesis on chromatic adaptation as a concep-
tual extension of Grassmann's laws of additive color mixture across two viewing conditions
[4]. Although chromatic adaptation affects the three types of cone responses differently, the
von Kries model implies that the relative spectral sensitivity of each of the three cone mecha-
nisms remains unchanged. According to this hypothesis, adaptation in the three cone types
is independent and inversely related to the cone responses to the adapting stimulus. The von
Kries hypothesis can be summarized in Eq (1), where LMS are the initial cone responses,

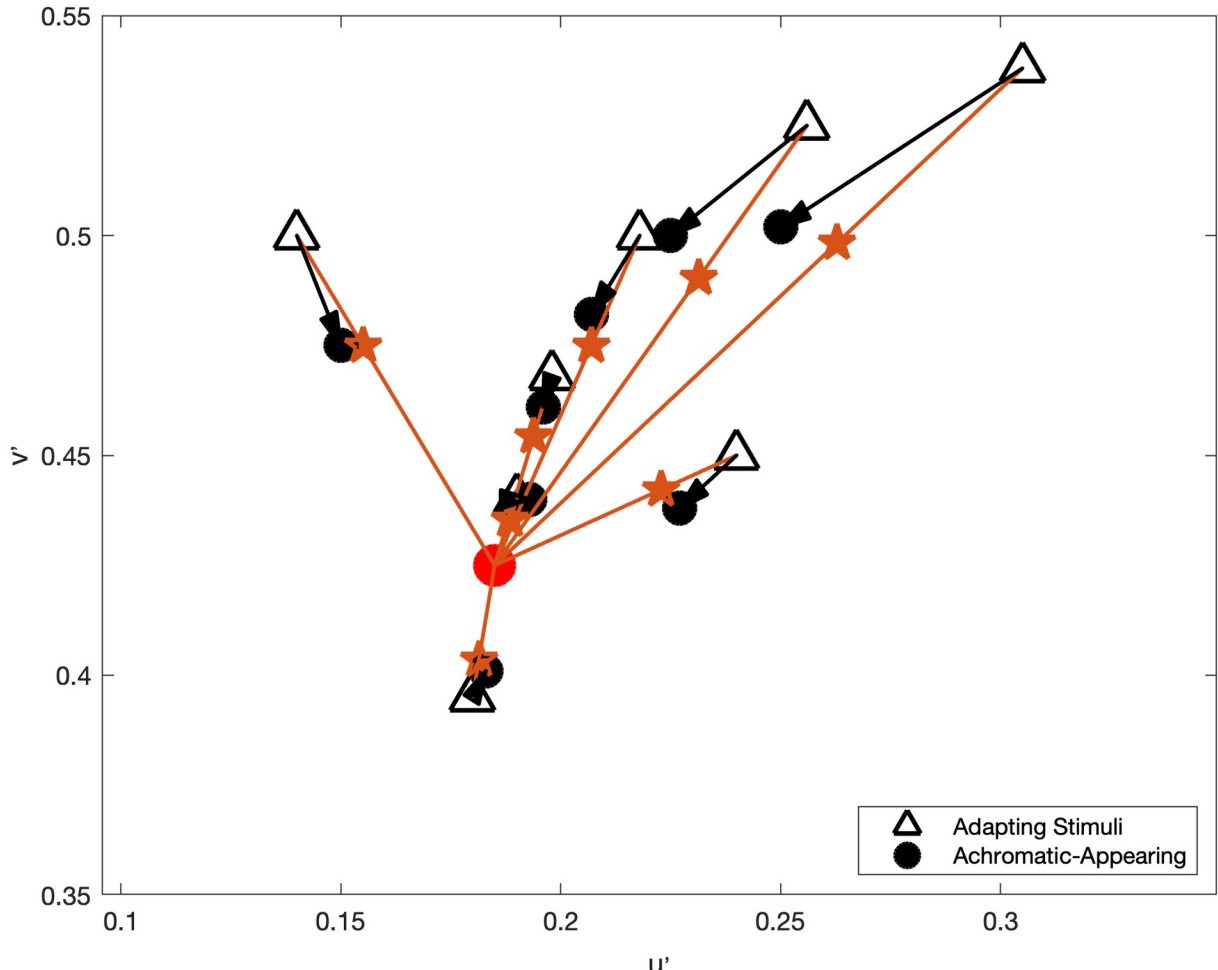

**Fig 1. The three computational steps included in a typical chromatic adaptation transform (CAT).**

$L_nM_nS_n$ are cone responses to the adapting stimulus, and $L_aM_aS_a$ are the post-adaptation cone signals [3].

$$\begin{bmatrix} L_a \\ M_a \\ S_a \end{bmatrix} = \begin{bmatrix} \frac{1}{L_n} & 0 & 0 \\ 0 & \frac{1}{M_n} & 0 \\ 0 & 0 & \frac{1}{S_n} \end{bmatrix} \begin{bmatrix} L \\ M \\ S \end{bmatrix} \qquad (1)$$

Chromatic adaptation is not always complete [5]. Therefore, in many adaptation models a D factor for the degree of adaptation has been developed to expand the von Kries model to account for incomplete chromatic adaptation. Such a D factor is embedded in most accurate general-purpose CAT models, such as CAT02 and CAT16. Eq (2) illustrates a CAT model accounting for incomplete chromatic adaptation.

$$\begin{bmatrix} L_a \\ M_a \\ S_a \end{bmatrix} = \begin{bmatrix} \frac{1}{DL_n + (1-D)L_r} & 0 & 0 \\ 0 & \frac{1}{DM_n + (1-D)M_r} & 0 \\ 0 & 0 & \frac{1}{DS_n + (1-D)S_r} \end{bmatrix} \begin{bmatrix} L \\ M \\ S \end{bmatrix} \qquad (2)$$

$L_rM_rS_r$ refer to the responses to the reference illuminant. In CAT02 and CAT16, the reference illuminant is the so-called equal-energy (EE) illuminant (L = M = S = 100). However, various data sets [5–7] illustrate that the line segment connecting the adapting stimulus to the achromatic-appearing stimulus does not project toward EE. In other words, EE as the reference point is not strictly valid. Fairchild proposed a new chromatic adaptation model (vk20) that takes sky blue (a Planckain radiator at 15000K) as a new reference point. vK20 relies on three chromaticities and three-D values. Besides, $L_rM_rS_r$ and $L_nM_nS_n$, the vK20 model includes the previous adapting illuminant $L_pM_pS_p$ in the model to account for hysteresis from other recently viewed. Stimuli [3]. The vK20 formulation is given in Eq (3), and the sum of $D_n$, $D_r$, and $D_p$ should be 1.0. As shown in Fig 2, all the line segments representing the direction of incomplete adaptation in a u'v' chromaticity diagram are approximately

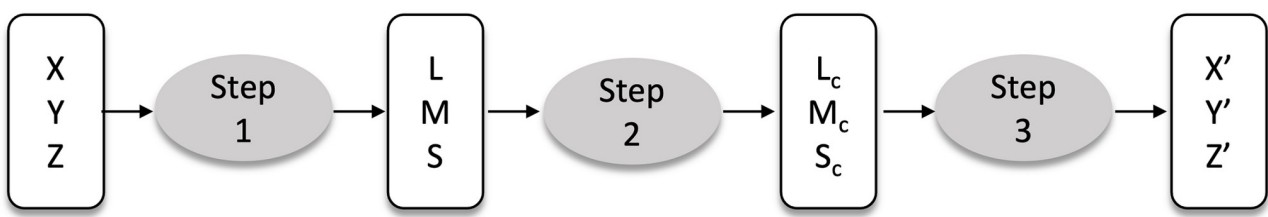

**Fig 2. The 15000K reference point is shown by a red circle, orange lines indicate projections from each adapting chromaticity to that point, and orange pentagrams reflect the prediction of 70% adaptation from that reference to adapting chromaticity.** This plot is recreated from Fairchild (2020).

**Table 1. Summary of all types of cone responses.**

| L M S | Cone responses to the stimulus under test illuminant. |
|---|---|
| Ln Mn Sn | Cone responses to test illuminant, or other selected adapting stimulus. |
| Lr Mr Sr | Cone responses to the reference illuminant. In CAM16 or CAM02, the reference illuminant is equal-energy (EE) illuminant ($L_r = M_r = S_r = 100$). In vK20 the reference illuminant is 15000K ($L_r = 95.41$, $M_r = 103.87$, $S_r = 169.81$). |
| $L_p$ $M_p$ $S_p$ | Cone responses to a previous adapting illuminant or other adapting stimulus. |
| $L_a$ $M_a$ $S_a$ | Post-adaptation cone signals |
| $L_c$ $M_c$ $S_c$ | Adapted cone responses. ($L_c = L_a {}^* L_r$, $M_c = M_a {}^* M_r$, $S_c = S_a {}^* S_r$) |
| L' M' S' | Adapted cone signal of adapting stimulus. Only mentioned in the WGM method. |

projected to sky blue(15000K).

$$\begin{bmatrix} L_a \\ M_a \\ S_a \end{bmatrix} = \begin{bmatrix} \dfrac{1}{D_nL_n + D_rL_r + D_pL_p} & 0 & 0 \\ 0 & \dfrac{1}{D_nM_n + D_rM_r + D_pM_p} & 0 \\ 0 & 0 & \dfrac{1}{D_nS_n + D_rS_r + D_pS_p} \end{bmatrix} \begin{bmatrix} L \\ M \\ S \end{bmatrix} \quad (3)$$

In order to better explain the terms used in this paper, the definitions of the different cone response are summarized in Table 1.

It is well known that all of these widely-used chromatic adaptation models (CAT02, CAT16, and vK20) are acutely based on the von Kries hypothesis, which constrains the direct prediction of incomplete adaptation along the line segment from adapting stimulus to a reference point in a chromaticity diagram such as CIE u'v'. However, not only the results from Fig 2 but also, for example, the data from Shen and Fairchild [6] and Zhai and Luo [7] suggest that the prediction of incomplete adaptation shouldn't follow the line segment connecting the adapting stimulus to the reference point. Instead, it should move along the curve like the Planckian or Daylight locus from adapting stimulus to reference point. Therefore, this paper proposes a better-performing, and more physiologically plausible, CAT model that can predict the degree incomplete adaptation moving along the curve in chromaticity by using the geometric mean method.

## Geometric mean method

Most chromatic adaptation models, both theoretically and mathematically, can be traced back to von Kries hypothesis and its more recent mathematical implementations. This paper explores whether we really need the normal linear von Kries hypothesis and if more recent and generalized neurophysiological theory might apply.

In a recent paper, Wong [8] illustrates an exciting concept that the adaptation in the peripheral sensory systems across many types of sensory adaptation, and species, is found to follow a simple mathematical relationship: the geometric mean. Wong's proposed relationship, illustrated in Fig 3, involves the steady-state spontaneous rate (SR) prior to the introduction of stimulus, the peak response to stimulus (PR), and the subsequent new steady-state response (SS) after adaptation. It is hypothesized that SS can be predicted as the geometric mean of SR and PR. The mathematical relationship is shown in Eq (4).

$$SS = \sqrt{PR \times SR} \quad (4)$$

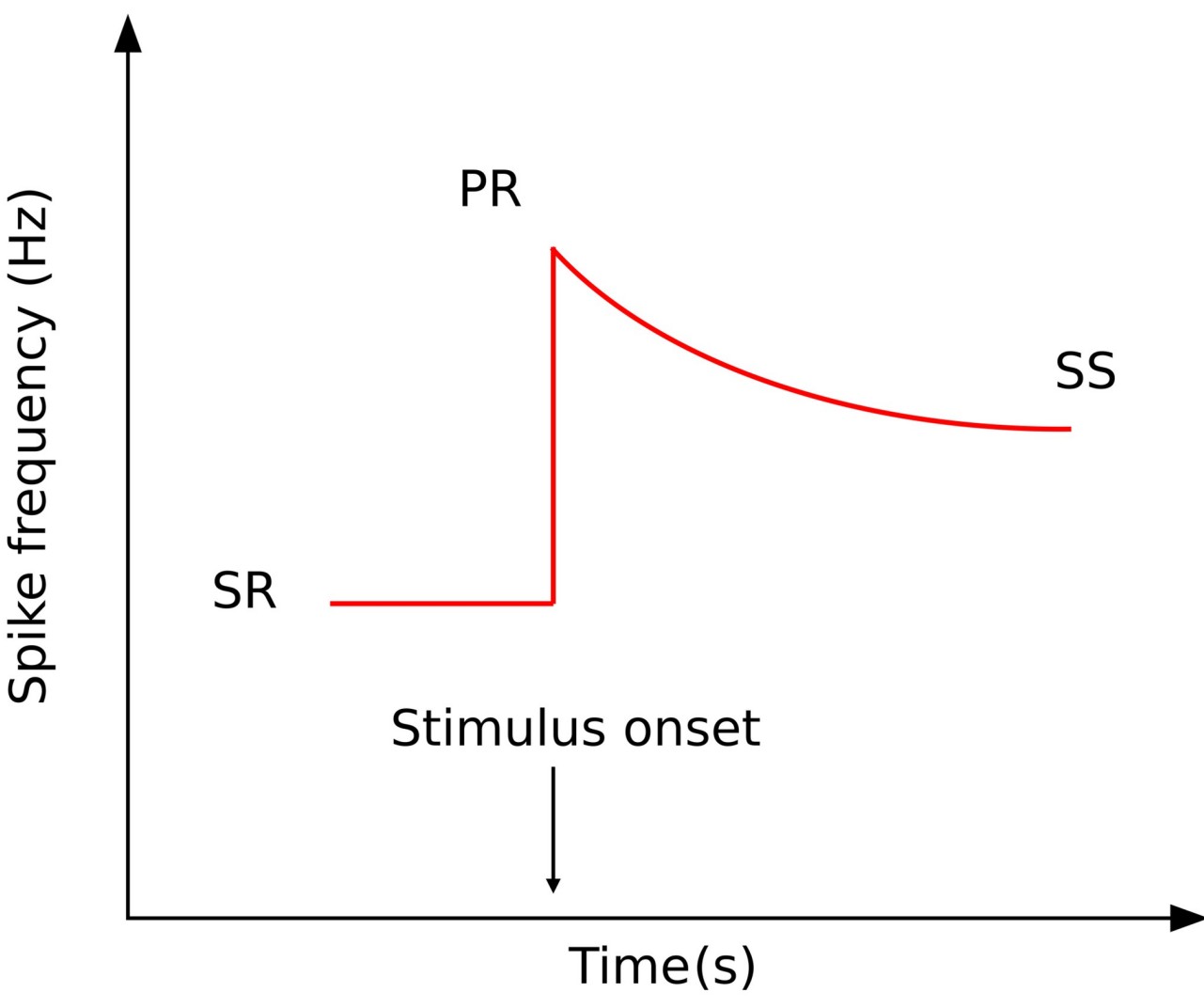

**Fig 3. Peripheral sensory adaptation curve (ideal situation).** This figure is recreated based on Fig 1 in Wong (2021).

In Wong's paper [8], more than 200 measurements taken from different types of sensory adaptation and different species are summarized and shown to be compatible with geometric mean model. Examples include: (1) Auditory response in guinea pig fiber, gerbil fiber, ferrets, and saccular nerve fibers of goldfish. (2) The responses from lateral line systems in fish. (3) Stretch responses in crayfish and frogs. (4) Response of olfactory receptor neurons in fruit flies. (5) Taste recording in fruit fly sensilla, caterpillars, and blowflies. (6) Response to cooling in beetles. (7) Vision data from On-Centre ganglion cells in the cat. Wong did not explore the applicability of the geometric mean model to chromatic adaptation in cone photoreceptors.

Though cone photoreceptors have continuous voltage responses rather than spike frequency, the geometric mean concept might still apply to cone responses directly (linear in the first stage). It is also likely that some portion of chromatic adaptation is occurring in neural cells beyond the cones that do have spike-frequency modulated responses such as those explored by Wong [8]. Therefore, in the proposed new CAT model, PR is the cone response to

adapting stimulus ($L_nM_nS_n$), SR is the cone response to reference point ($L_rM_rS_r$), and the SS is the adapted cone signal of adapting stimulus (L'M'S'). See Eq (5). The neutral-appearing chromatic adaptation results of Fairchild [9], along with the predictions using the geometric mean method are shown in Fig 4. It is surprising and encouraging that such a simple model potentially works so well, and that the prediction results (red dots in Fig 4) move along the desired curve (either daylight curve or Planckian locus) rather than a straight line. Note that in this geometric mean model there is no factor for degree of adaptation; the geometric mean automatically computes appropriate degree of adaptation. This is the first time that a simple adaptation model, with no arbitrary or empirical degree of adaptation factor, has been capable of predicting the observed degree of incomplete sensory chromatic adaptation first quantified in the late 1980s and early 1990s.

$$L'M'S' = \sqrt{L_nM_nS_n \times L_rM_rS_r} \tag{5}$$

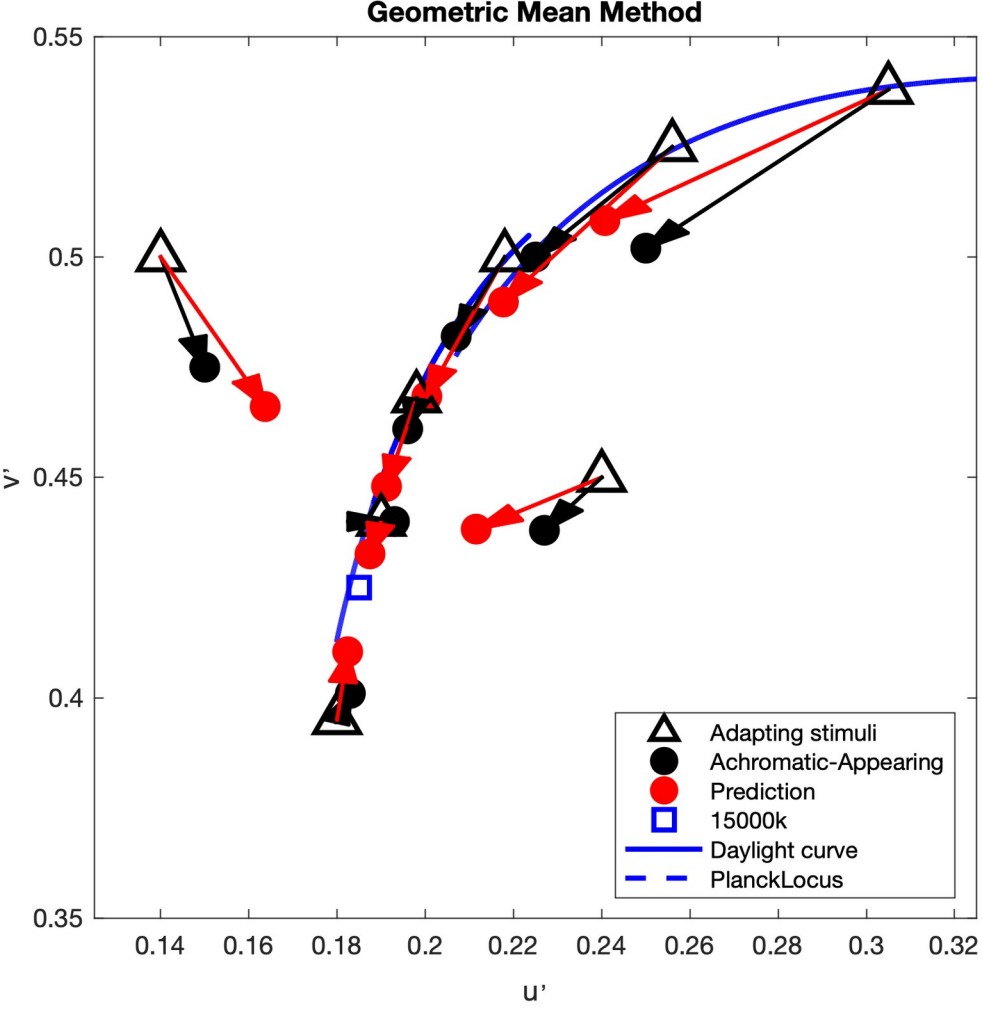

Fig 4. The blue square represents the reference point (15000K). Red arrows indicate projection from each adapting chromaticity to prediction results by the geometric mean method. The blue dotted line represents the Planckian locus, while the blue line indicates the daylight locus.

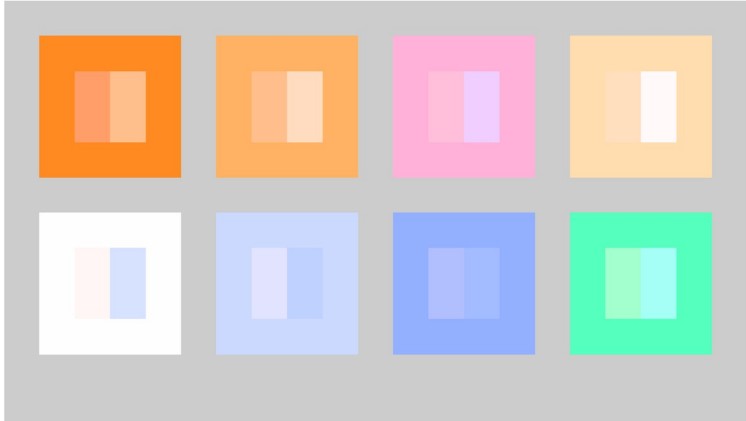

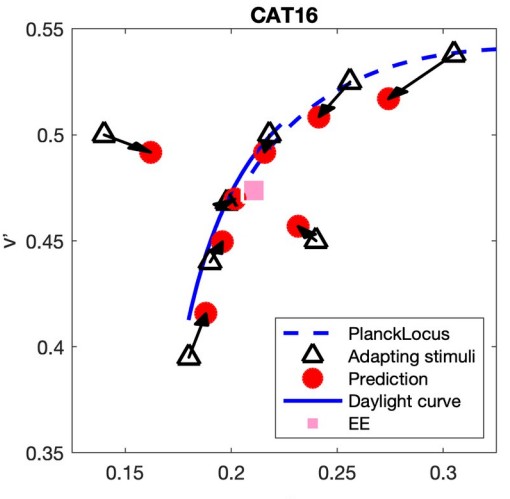 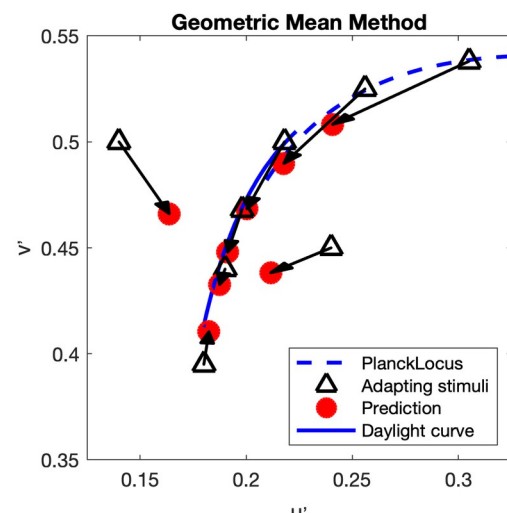

**Fig 5.** Top: The perceptual neutral point predicted by CAT16 with D = 0.7 (the rectangle on the left inside each square) and geometric mean method (the rectangle on the right inside each square. The color of each square represents the adapting stimuli. Bottom: Red arrows indicate projection from each adapting chromaticity (black triangle) to predicted neutral-appearing results (red circle) by CAT16 with D = 0.7 (left) and the geometric mean method (right).

In order to better visualize the shift of neutral point across different ambient illumination chromaticities, a comparison of the geometric mean method with CAT16 (D = 0.7) predictions is shown in Fig 5 (top). The colors of the adapting stimuli are represented by each square. The rectangle on the left inside each square is the neutral point predicated by CAT16 (D = 0.7) while the one on the right is the neutral point predicated by geometric mean method. Chromaticity values of the predictions and adapting stimuli are shown in the bottom of Fig 5 (left: CAT16 with D = 0.7, right: geometric mean) This figure (both top and bottom) shows the geometric mean method approximately following the Planckian or daylight locus (color shift from yellowish to blueish and move along Planckian locus) and CAT with D = 0.7 trending toward to EE (more pinkish).

## Weighted Geometric Mean (WGM) method

Like all CAT models, if necessary, a D factor for degree adaptation can be developed to expand the geometric mean method to account for more complete chromatic adaptation and differences in chromatic adaptation for various viewing conditions when cognitive adaptation mechanisms are active to various degrees in addition to the simple sensory mechanisms that are well predicted by the geometric mean alone. Therefore, Eq (5) can be extended and rewritten as Eq (6):

$$\begin{bmatrix} L' \\ M' \\ S' \end{bmatrix} = \begin{bmatrix} L_n \\ M_n \\ S_n \end{bmatrix}^{D} \cdot \times \begin{bmatrix} L_r \\ M_r \\ S_r \end{bmatrix}^{(1-D)} \tag{6}$$

When D is equal to 1, complete adaptation to the adapting stimulus is predicted (sometimes referred to as "discounting the illuminant", which is a cognitive process), when D is equal to 0 the model predicts adaptation to the reference condition (which might happen for briefly flashed stimuli), and intermediate values of D predict various degrees of adaptation in between. When D is equal to 0.5, the weighted geometric mean method is identical with the simple geometric mean method described by Eq (5). To check the performance of the WGM model, three CAT models (CAT16, vK20, and WGM) were selected to predict the psychophysical experimental data of Fairchild [9]. The MATLAB fmincon function was used to optimize the D value to minimize color difference (Euclidean distance in u'v' diagram) between prediction and experiment results. Optimized D values were computed both individually and totally (single D value) for each adapting stimulus. The optimized results (both individual D values for each adapting stimulus and single D value for each model) are illustrated in Fig 6.

It is evident that the reference point will impact the model performance since it is the only difference between CAT16 and vK20 in this calculation ($D_p = 0$ is assumed in vK20). In the WGM model, a 15000K reference point was also chosen based on previous results. However, another optimization method was completed by fitting both the D value (also individually for each adapting stimulus and singly by model) as well as the reference point to minimize the color difference between prediction and experiment results. In Fig 6, WGM(CCT) and WGM (CCT, single) represent the results with reference point optimization. All the optimized D values are summarized in Table 2. The Box plot in terms of chromaticity difference is shown in Fig 6 (bottom right) indicates the prediction results of single D value in CAT16 is very large. In other words, vK20 and WGM perform better than CAT16 with single D values. This is an effective illustration of the theoretical error in CAT16, which holds that D value, as modeled in CIECAM16, is only a function of luminance level. As matter of fact, for the CIECAM16 formulation, optimal D values also depend on the chromaticity value of adapting stimuli and perhaps other factors. The incorrect reference illuminant (EE) is also a factor in this poor performance. WGM and vK20 do not have these theoretical flaws. In addition, the best model performance is also exhibited by WGM with individually optimized D values.

## Do we still need von Kries?

Yes. Though achromatic appearance can be more accurately predicted by this modern neurophysiological theory (WGM), the results cannot be directly correlated to $L_aM_aS_a$ (post-adaptation cone signals) for arbitrary stimulus colors. In other words, chromatic corresponding-colors under different illumination conditions can't be predicted, but the relevant neutral point is defined by the model. Implementation of the WGM model should also be based on the von Kries hypothesis, or modified from von Kries model, with the understanding that the

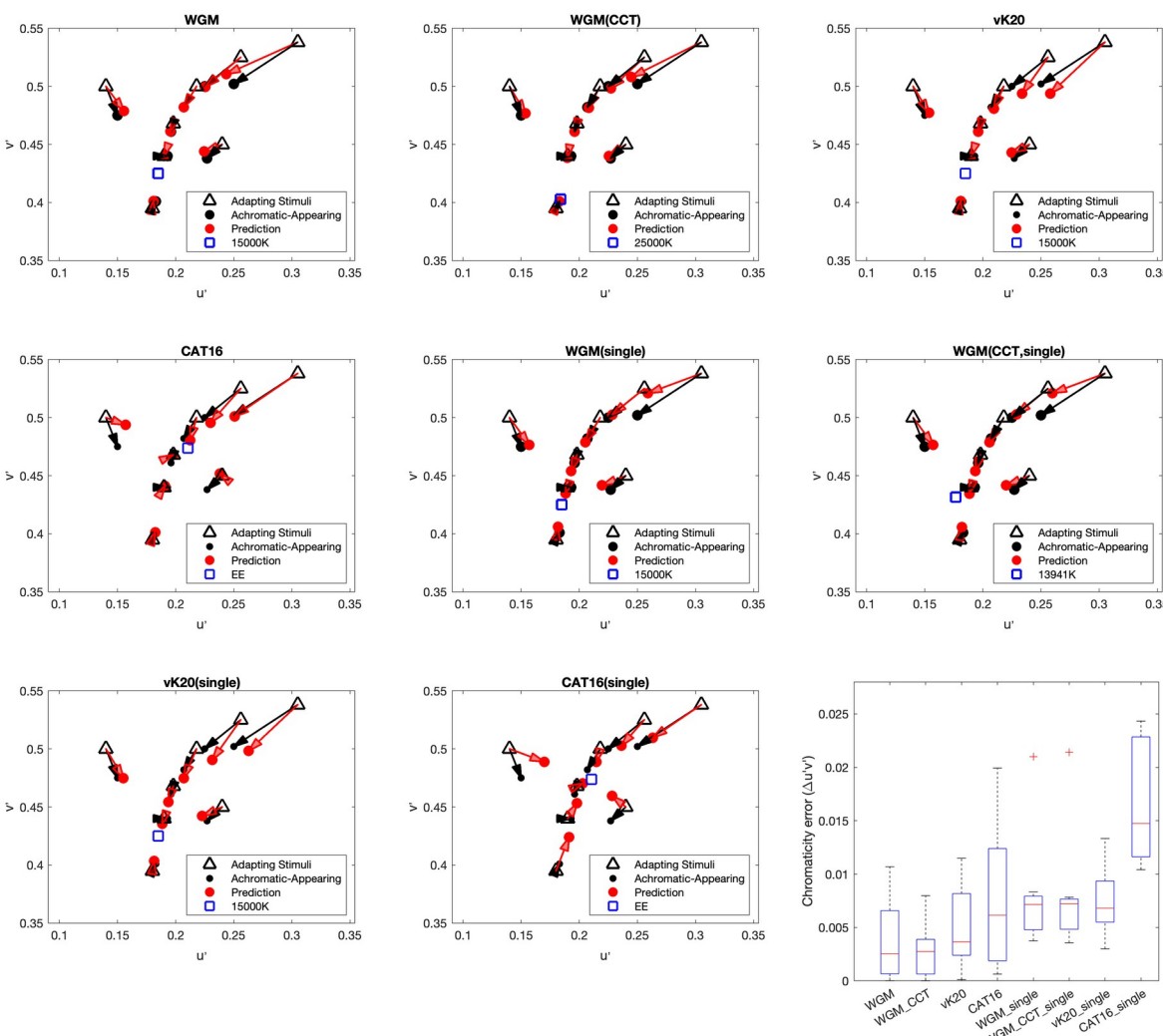

**Fig 6. The perceptual neutral point predicted by CAT16 with D = 0.7 (the rectangle on the left inside each square) and geometric mean method (the rectangle on the right inside each square).** The color of each square represents the adapting stimuli. Bottom: Red arrows indicate projection from each adapting chromaticity (black triangle) to predicted neutral-appearing results (red circle) by CAT16 with D = 0.7 (left) and the geometric mean method (right).

**Table 2. Optimized D values for CAT16 and WGM.**

| Adapting stimuli | | | | | | | | |
|---|---|---|---|---|---|---|---|---|
| CAT16 | 0.4563 | 0.4526 | 0.9274 | 0.2677 | 1 | 0.9575 | 0.9066 | 0.7692 |
| CAT16 (single) | 0.5896 | | | | | | | |
| vK20 | 0.6619 | 0.7310 | 0.7333 | 0.7745 | 0.8506 | 1 | 0.7768 | 0.7293 |
| vK20 (single) | 0.7332 | | | | | | | |
| WGM | 0.5217 | 0.6067 | 0.7324 | 0.6944 | 0.8213 | 1 | 0.7989 | 0.6752 |
| WGM (single) | 0.6446 | | | | | | | |
| WGM (reference point:25000K) | 0.5421 | 0.6406 | 0.7595 | 0.7424 | 0.8728 | 0.9541 | 0.2276 | 0.7141 |
| WGM (single, reference point:13941K) | 0.6474 | | | | | | | |

adapting signals are defined by geometrical combinations of the adapting and reference stimuli rather than arithmetical combinations. Therefore, by incorporating the geometric mean computation as the incomplete adaptation equation, the final WGM model is shown in Eq (7):

$$
\begin{bmatrix} L_a \\ M_a \\ S_a \end{bmatrix} = \begin{bmatrix} \dfrac{1}{L_n^D \times L_r^{1-D}} & 0 & 0 \\ 0 & \dfrac{1}{M_n^D \times M_r^{1-D}} & 0 \\ 0 & 0 & \dfrac{1}{S_n^D \times S_r^{1-D}} \end{bmatrix} \begin{bmatrix} L \\ M \\ S \end{bmatrix} \tag{7}
$$

## Testing of the WGM model

The WGM model was tested by using four different corresponding-colors data sets. The first set (Derby/Leeds data set) includes fourteen sub-data sets collected from nine sources: the Color Science Association of Japan (CSAJ), Helson, Lam and Rigg, LUTCHI, Kuo and Luo, Breneman, Braun and Fairchild, and McCann [10]. Table 1 from the study by Luo and Rhodes [10] summarizes the experimental settings for the Derby/Leeds data set.

The second set (Fairchild Reversibility data [11]) came from an experiment in which the observers select neutral color (self-luminance color stimuli) under the background of CIE Illuminant D65 and CIE Illuminant A. In order to produce precise color stimuli, an Eizo Color Edge CG279x calibrated display was employed. The mean luminance of the adapting background was 400 cd/m² for the D65 and 241 cd/m² for the illuminant A background. This study focused on sensory chromatic adaptation as the stimuli could not be perceived in such a way as to allow for cognitive discounting of the illuminant.

The third set, the Cai data [12], came from the short-term memory matching experiment. Observers remembered the color appearance of test colors (physical sample) which presented under simulated illuminant A after fully adaptation. Then, the light sources switched to simulated D65 and the observer re-adapted to this lighting and choose the best matches based on their memory.

The fourth set (Shen data [6]) accumulated from the experiment in which the observers select neutral color stimuli (physical sample) under illuminant D65 and A. Three luminance levels (100,200,300 cd/m²) of ambient light, both D65 and A, were used in this experiment.

The results of testing three different models (CAT16, vK20 and WGM (15000K)) using the above-mentioned data sets are shown in Table 3. The testing method is based on the concept that the adapted cone responses of corresponding color are the same under adaptation to the reference point, as illustrated in Fig 7. In Fig 7, the points A and B represent original tristimulus value (measured directly from spectroradiometer) for a pair of corresponding color in u'v' diagram. A' and B' represent the perceived color (after adaptation) under reference illuminant (i.e., EE in CAT16 or 15000K in WGM) by using any CAT transform. A' was transformed from A while B' was transformed from B. The point A' and B' should locate at the same

**Table 3. Mean chromaticity error (Δu'v') of predictions from experimental corresponding-color data for CAT16, vK20 and WGM.**

| Model | Derby/Leeds data | Cai data | Fairchild Reversibility Data (achromatic appearing) | Shen Data |
|-------|------------------|----------|------------------------------------------------------|-----------|
| CAT16 | 0.0104 | 0.0261 | 0.0180 | 0.0087 |
| vK20 | 0.0116 | 0.0259 | 0.0100 | 0.0039 |
| WGM | 0.0107 | 0.0263 | 0.0085 | 0.0042 |

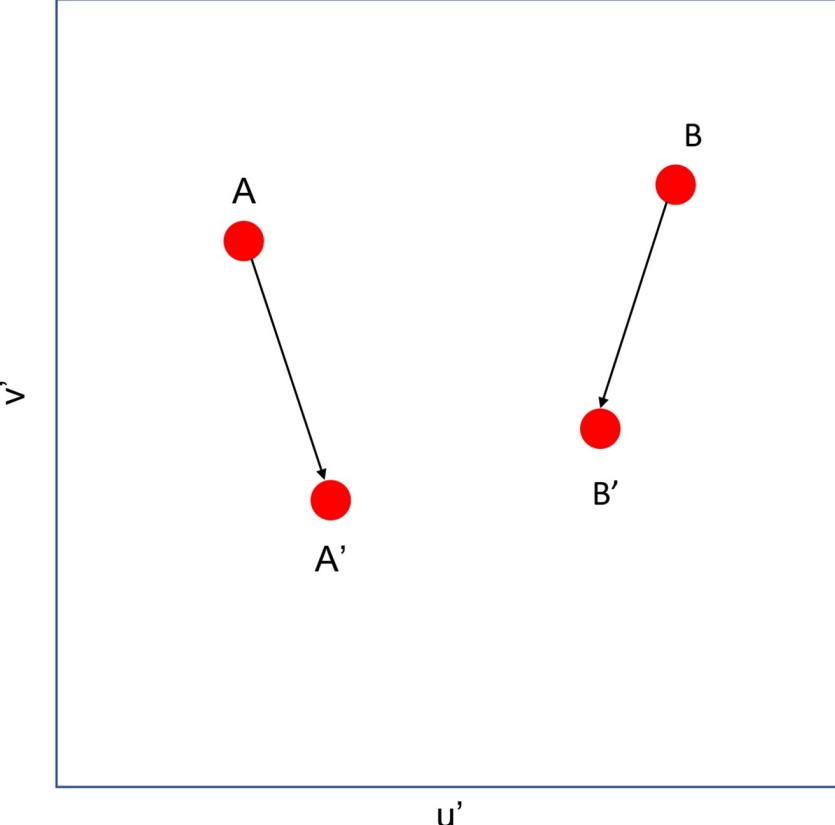

**Fig 7. Quantitive analysis of the performance (distance between A'B') of any CAT model.** Points A and B represents a pair of corresponding color located at u'v' diagram. A' and B' represents the model prediction results (corresponding colors for the reference illumination) by any CAT model.

position if there is perfect agreement between visual assessment and model prediction. The distance between A' and B' represents the magnitude error of test model of this one pair of corresponding color. The mean of the distance between A' and B' from corresponding data sets was used to measure the overall performance of each model.

The test procedures of the Derby/Leeds data set and the Cai data are as follow:

- Calculate the adapted cone response ($L_cM_cS_c$_1) of corresponding data under first illumination using different CAT.

- Calculate the adapted cone response ($L_cM_cS_c$_2) of corresponding data under second illumination using different CAT.

- Convert adapted cone response ($L_cM_cS_c$_1 and $L_cM_cS_c$_2) into CIE 1976 u'v' (u'v'Y_1 and u'v'Y_2).

- Optimize the D value to minimize the distance ($\Delta$u'v') between u'v'Y_1 and u'v'Y_2.

- Calculate the mean distance value from corresponding data set.

The Fairchild Reversibility data and Shen data are both selected achromatic-appearing (neutral) color under different lighting conditions. In such cases, the test procedures of those neutral colors follow the same method as testing the psychophysical experimental data of

Fairchild 1990 data [9] (shown in Fig 6). The performance between different CATs in fitting neutral color data sets is also shown in Table 3.

Kuo and Luo [13] discussed the magnitude of observer variation in studying chromatic adaptation. They concluded that a satisfactory CAT should have a prediction error of four CMC (1:1) units or less (roughly 4x the magnitude of a Just Noticeable Difference (JND) for simple color stimuli with no change in adaptation). The Derby/Leeds data was recommended by the CIE as the benchmark to test CAT models [10]. Since the three CAT models' predictions from the Derby/Leeds data were all less than 4 JND (1 JND is roughly equal to 0.004 u'v' [14] so 4 JND would be about 0.016 u'v'), they are all regarded as excellent chromatic adaptation models according to these CIE criteria. ANOVA analyses were performed to determine whether the differences in predicting corresponding-color data of each model were statistically significant. The ANOVA results of Derby/Leeds data shows the CAT16 is significantly better (F = 3.93, Prob>F: 0.0197) than vK20 and there are no significant differences between WGM and other two CAT models. For the Cai data, CAT16, vK20, and WGM produced results that were not significantly different from one another.

However, the accuracy of CAT16 and WGM can't be separated from the Derby/Leeds and the Cai data. This is because the observer variation in collecting corresponding color data sets was larger than accuracy variation between these two CAT models. The Derby/Leeds data were accumulated from three different methods (Haploscopic match, Memory match, Magnitude estimation) while the Cai data were only based on Memory matching. There is unavoidable limitation existing in both the Derby/Leeds and Cai data sets, such as the color memory issue or errors from cognitive mechanisms such as inaccurate color memory [4]. Hence, the data sets are not precise enough to distinguish the difference between CAT16 and WGM. In 1981, Wright suggested that selecting the stimulus that appeared a neutral grey at various adapting color temperatures should potentially be a very powerful method for study chromatic adaptation because grey is a particularly unambiguous color to memorize [15]. Therefore, achromatic corresponding data sets were also chosen (Fairchild Reversibility and Shen data) to test the models. Examination of the results for both the Fairchild Reversibility data and Shen data illustrate that the CAT16 predictions are significantly worse (F = 69.85 Prob>F: <0.0001 for the Fairchild Reversibility data. F = 76.52 Prob>F: <0.0001 for the Shen data) than vK20 and WGM. There are no significant differences between the vk20 predictions and the WGM predictions for these data sets, but the WGM model can be preferred on theoretical grounds. Observer variability for each dataset is shown as violin plot in Fig 8.

In addition, unique hue could also be a suitable choice for creating corresponding color. For example, observers select unique hues with fixed lightness value under D65 and illuminant A respectively. These pairs of unique hues could be considered as corresponding color for the hue dimension only as possible changes in perceived saturation are not accounted for. There are also unique hue data sets which included in Fairchild's research [11]. Statistical tests (not included in Table3) show that there are no significant differences in predicting the unique hue data sets from above mentioned models. Although the lack of observers in that study may have contributed to the inability to distinguish between the three models' performances, it is still important to note that the unique hue settings exhibit substantial inter-observer variations, which has been shown to be insufficient for measuring observer metamerism as an intrinsic reference [16]. In order to test whether selecting unique hue stimulus under different illumination can serve as useful tool to collecting corresponding data, more unique hue corresponding-colors need to be accumulated and tested in the future.

Finally, it is worth to ask the question of what a perfect CAT model would be. On the one hand, a perfect CAT model should predict corresponding colors accurately. Based on our statistical test results, WGM outperformed the other two CATs in general, especially for the data

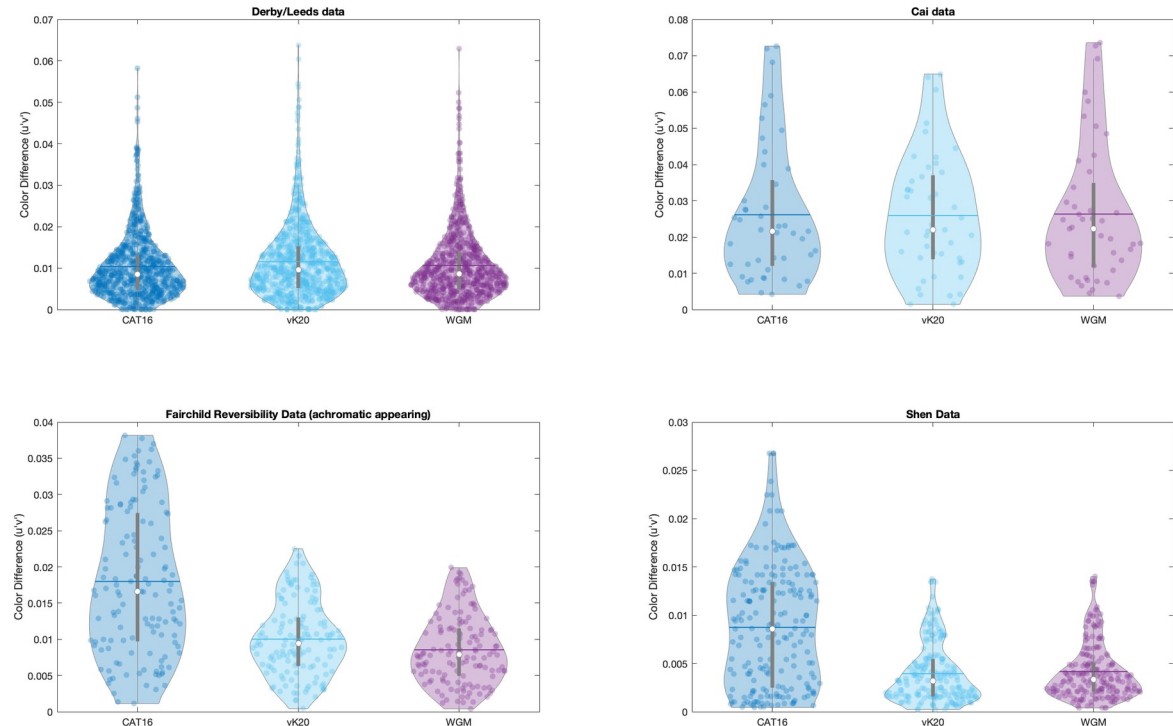

**Fig 8. Violin plots of the observer variability for four chosen corresponding data sets.**

of achromatic memory matching. On the other hand, a perfect CAT is also able to explore the functioning of the visual mechanism. There is no physiological justification for either the methods used in either CAT16 or vK20 to predict incomplete chromatic adaptation. However, the theoretical foundation of WGM is the geometric mean approach, which has been demonstrated to be physiologically plausible across a wide range of animal species and modalities of sensory adaptation. By adding the weighting factor (creating WGM), the geometric mean equation can also anticipate the impact of various cognitive mechanisms of chromatic adaptation, which is also physiologically and psychologically reasonable.

## Conclusion

Unlike any other von Kries-type chromatic adaptation models, WGM is able to intrinsically predict incomplete adaptation moving along the Planckian or daylight loci in chromaticity and to automatically compute an appropriate degree of incomplete sensory chromatic adaptation with no need for empirical adaptation factors. Degree of adaptation factors, however, will be needed in practical applications where both sensory and cognitive adaptation mechanisms are active [4]. The determination of degree of adaptation remains an open problem, as the D value is not only influenced by ambient light intensity, which is incorporated in modern color appearance models, but also by other factors such as the chromaticity value of adapting illumination. Obtaining more reliable corresponding data would be valuable in constructing a robust model capable of accurately predicting the D value.

The geometric mean method has been shown to generalize across sensory modality and to hold across a wide range of animal species, which is the theoretical basis of WGM. The analyses in this paper illustrate that the geometric mean method also applies to the prediction of

sensory chromatic adaptation in human observers. This model was tested with data from four different corresponding-colors data sets and shown to perform significantly better than CAT16 and vK20. The WGM model is also much simpler, more physiologically plausible, and easier to implement. Our future work will involve collecting more corresponding data, considering both inter and intra observer variation. This will enable us to conduct statistical tests on different models that take observers' variability into account. Part of results and analyses of this paper was published and presented at the IS&T 30th Color and Imaging Conference [17].

## Author Contributions

**Conceptualization:** Che Shen, Mark D. Fairchild.

**Data curation:** Che Shen, Mark D. Fairchild.

**Formal analysis:** Che Shen, Mark D. Fairchild.

**Funding acquisition:** Mark D. Fairchild.

**Investigation:** Che Shen, Mark D. Fairchild.

**Methodology:** Che Shen, Mark D. Fairchild.

**Project administration:** Che Shen, Mark D. Fairchild.

**Resources:** Che Shen, Mark D. Fairchild.

**Software:** Che Shen.

**Supervision:** Che Shen, Mark D. Fairchild.

**Validation:** Che Shen, Mark D. Fairchild.

**Visualization:** Che Shen, Mark D. Fairchild.

**Writing – original draft:** Che Shen.

**Writing – review & editing:** Che Shen, Mark D. Fairchild.

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
