## [Decision Letter · Decision Letter 0]

27 Mar 2023

PONE-D-23-01309Weighted Geometric Mean (WGM) method: A new chromatic adaptation modelPLOS ONE

Dear Dr. Shen,

Thank you for submitting your manuscript to PLOS ONE. After careful consideration, we feel that it has merit but does not fully meet PLOS ONE’s publication criteria as it currently stands. Therefore, we invite you to submit a revised version of the manuscript that addresses the points raised during the review process. Please note that the reviewer required more appropriate statistical analysis of the data and quality of data processing is one of the necessary publication criteria in PLoS ONE.I am sure you will be able to implement the changes and analysis required by the reviewer in due time so I encourage resubmission. 

We look forward to receiving your revised manuscript.

Kind regards,

J Malo

Academic Editor

PLOS ONE

2. We noted in your submission details that a portion of your manuscript may have been presented or published elsewhere. [Yes, at IS&T color image conference (30). Conference paper is part of our research.] Please clarify whether this conference proceeding was peer-reviewed and formally published. If this work was previously peer-reviewed and published, in the cover letter please provide the reason that this work does not constitute dual publication and should be included in the current manuscript.

3. We note that you have referenced (11.Fairchild MD. Reversibility of Corresponding Colors in Sensory Chromatic Adaptation. Unpublished Compete Report. Available from: http://markfairchild.org/PDFs/PRO54.pdf.) which has currently not yet been accepted for publication. Please remove this from your References and amend this to state in the body of your manuscript: (ie “Bewick et al. [Unpublished]”) as detailed online in our guide for authors

Reviewers' comments:

Reviewer's Responses to Questions

**Comments to the Author**

1. Is the manuscript technically sound, and do the data support the conclusions?

Reviewer #1: Partly

2. Has the statistical analysis been performed appropriately and rigorously? 

Reviewer #1: Yes

3. Have the authors made all data underlying the findings in their manuscript fully available?

Reviewer #1: Yes

4. Is the manuscript presented in an intelligible fashion and written in standard English?

Reviewer #1: Yes

5. Review Comments to the Author

Reviewer #1: The paper proposes a new CAT based on using the Weighted Geometric Mean. The problem is interesting in the color science field and the work provides insight to the matter. The paper is well written and is technically sound. My only concern is related to the datasets used in the experimental evaluation because I believe that data uncertainty (observer's variability) of the data sets should be provided (include in table 3). Other comments:

1. I think a couple of more sentences in the abstract can be helpful for contextualizing the problem better.

2. Please include Von Kries reference on line 25 of page 1

3. The D value in the adaptation has been set by optimization. Please detail what dataset has been used to do this and whether the found value is good for other datasets.

4. Can D be set just this way of is setting D an open problem?

5. Include observer's responses variability in table 3 for each dataset. This is important to contextualize the differences between the methods to the observer's variability, which is helpful in understanding how different the methods perform

6. If there was a way to make statistical tests taking into account oberservers variablity, it would be great.

6. PLOS authors have the option to publish the peer review history of their article (what does this mean?). If published, this will include your full peer review and any attached files.

Reviewer #1: No

---

## [Author Response · Author response to Decision Letter 0]

11 May 2023

Dear Editor,

Thank you for considering our manuscript Weighted Geometric Mean (WGM) method: A new chromatic adaptation model for publication in PLOS ONE. We appreciate the time and effort that you and the reviewers have taken to provide valuable feedback on our work.

We have carefully considered the points raised by the academic editor and reviewer(s) and would like to respond to each of them as follows:

From reviewer(s): 

Point 1: I think a couple of more sentences in the abstract can be helpful for contextualizing the problem better.

Our Response: Some sentences added in the abstract.

Point 2: Please include Von Kries reference on line 25 of page 1.

Our Response: Included

Point 3: The D value in the adaptation has been set by optimization. Please detail what dataset has been used to do this and whether the found value is good for other datasets.

Our Response: The D- value is highly depended on various view condition such as ambient light intensity and correlated color temperature (CCT), etc. Due to the absence of robust model in predicting D value, we choose to optimize it to best fit the experiment data for each model. Therefore, comparing each CAT model is only related to model itself rather than the method setting D value. 

Point 4: Can D be set just this way of is setting D an open problem?

Our Response: As the mentioned in point 3, this D-value setting method enables us to compare model performance instead of the methodology in setting D value, which remains an open problem in color science arear. The equation used to set the D-value in the widely-used CAT model, CAT16, only takes into account the ambient light intensity and surround condition. However, there may be other factors that could potentially affect D-value such as the chromaticity value of ambient light. 

Here are some references in discussing setting D: 

Smet, Kevin AG, et al. "Study of chromatic adaptation using memory color matches, Part II: colored illuminants." Optics express 25.7 (2017): 8350-8365.

Zhai, Qiyan, and Ming R. Luo. "Study of chromatic adaptation via neutral white matches on different viewing media." Optics express 26.6 (2018): 7724-7739.

Point 5: Include observer's responses variability in table 3 for each dataset. This is important to contextualize the differences between the methods to the observer's variability, which is helpful in understanding how different the methods perform.

Our Response: Included the violin plot as figure 8 which shows the observer response variability in table 3 for each dataset.

Point 6: If there was a way to make statistical tests taking into account observers variablity, it would be great.

Our Response: Thanks for the comments. However, there might not be a good way to make the statistical test since some datasets are only based on few observers' data (Fairchild reversibility and Shen data) while Derby/Leeds data only shows the observer's average results. 

We have made the necessary revisions and amendments to our manuscript based on the comments and suggestions provided. We hope that our responses and revisions address all concerns and criticisms raised and demonstrate our commitment to improving the quality and clarity of our research.

Thank you again for your time and consideration.

Sincerely

Che Shen and Mark Fairchild

---

## [Decision Letter · Decision Letter 1]

29 Jun 2023

PONE-D-23-01309R1Weighted Geometric Mean (WGM) method: A new chromatic adaptation modelPLOS ONE

Dear Dr. Shen,

Thank you for submitting your revised manuscript to PLOS ONE. Before final acceptation, in order to properly put this research in context and stress the issues yet to be solved, it is mandatory to include in the conclusion section some comments on the points mentioned by the reviewer.  Please submit your revised manuscript by Aug 13 2023 11:59PM. If you will need more time than this to complete your revisions, please reply to this message or contact the journal office at plosone@plos.org. Please include the following items when submitting your revised manuscript:A rebuttal letter that responds to each point raised by the academic editor and reviewer(s). You should upload this letter as a separate file labeled 'Response to Reviewers'.A marked-up copy of your manuscript that highlights changes made to the original version. You should upload this as a separate file labeled 'Revised Manuscript with Track Changes'.An unmarked version of your revised paper without tracked changes. You should upload this as a separate file labeled 'Manuscript'.If applicable, we recommend that you deposit your laboratory protocols in protocols.io to enhance the reproducibility of your results. Protocols.io assigns your protocol its own identifier (DOI) so that it can be cited independently in the future. For instructions see: https://journals.plos.org/plosone/s/submission-guidelines#loc-laboratory-protocols. Additionally, PLOS ONE offers an option for publishing peer-reviewed Lab Protocol articles, which describe protocols hosted on protocols.io. Read more information on sharing protocols at https://plos.org/protocols?utm_medium=editorial-email&utm_source=authorletters&utm_campaign=protocols.

We look forward to receiving your revised manuscript.

Kind regards,

J Malo

Academic Editor

PLOS ONE

Journal Requirements:

Additional Editor Comments:

Please include comments in the conclusion section on the issues yet to be explored mentioned by the reviewer.

Reviewers' comments:

Reviewer's Responses to Questions

**Comments to the Author**

1. If the authors have adequately addressed your comments raised in a previous round of review and you feel that this manuscript is now acceptable for publication, you may indicate that here to bypass the “Comments to the Author” section, enter your conflict of interest statement in the “Confidential to Editor” section, and submit your "Accept" recommendation.

Reviewer #1: All comments have been addressed

2. Is the manuscript technically sound, and do the data support the conclusions?

Reviewer #1: Yes

3. Has the statistical analysis been performed appropriately and rigorously? 

Reviewer #1: Yes

4. Have the authors made all data underlying the findings in their manuscript fully available?

Reviewer #1: Yes

5. Is the manuscript presented in an intelligible fashion and written in standard English?

Reviewer #1: Yes

6. Review Comments to the Author

Reviewer #1: The authors have addressed my comments properly and I would just like them to include some future work along with the conclusions pointing out that:

1. The setting of D is an open problem of relevance in the field

2. It would be desiderable to have more data related to point 6 in the review

3. It would be interesting to research on what statistical tests that take into account observers responses variability should be used when comparint two CAT models.

7. PLOS authors have the option to publish the peer review history of their article (what does this mean?). If published, this will include your full peer review and any attached files.

Reviewer #1: No

---

## [Author Response · Author response to Decision Letter 1]

9 Jul 2023

From reviewer(s): 

Point 1: The authors have addressed my comments properly and I would just like them to include some future work along with the conclusions pointing out that:

1. The setting of D is an open problem of relevance in the field

2. It would be desiderable to have more data related to point 6 in the review

3. It would be interesting to research on what statistical tests that take into account observers responses variability should be used when comparint two CAT models

Our Response: Thanks for the suggestion. We included those points as part of our future work in the conclusion section.

---

## [Editor Report · Decision Letter 2]

1 Aug 2023

Weighted Geometric Mean (WGM) method: A new chromatic adaptation model

PONE-D-23-01309R2

Dear Dr. Shen,

We’re pleased to inform you that your manuscript has been judged scientifically suitable for publication and will be formally accepted for publication once it meets all outstanding technical requirements.

Kind regards,

J Malo

Academic Editor

PLOS ONE

Additional Editor Comments (optional): In this 2nd revision I think you addressed the last comments of the reviewer so I accept the paper in its current form. Nice work on chromatic adaptation!
---

## [Editor Report · Acceptance letter]

3 Aug 2023

PONE-D-23-01309R2 

Weighted Geometric Mean (WGM) method: A new chromatic adaptation model 

Dear Dr. Shen:

I'm pleased to inform you that your manuscript has been deemed suitable for publication in PLOS ONE. Congratulations! Your manuscript is now with our production department. 

Kind regards, 

on behalf of

Dr. J Malo 

Academic Editor

PLOS ONE